# Selenium-Containing Nanoformulations Capable of Alleviating Abiotic Stress in Plants

**DOI:** 10.3390/ijms26041697

**Published:** 2025-02-17

**Authors:** Olga Tsivileva

**Affiliations:** Institute of Biochemistry and Physiology of Plants and Microorganisms, Saratov Scientific Centre of the Russian Academy of Sciences, 13 Prospekt Entuziastov, Saratov 410049, Russia; tsivileva_o@ibppm.ru; Tel.: +7-845-297-0444

**Keywords:** selenium (Se), Se nanomaterials, Se nanoparticles (SeNPs), exogenous substances, cultured plants, abiotic stress, plant metabolites, reactive oxygen species (ROS), enzymatic antioxidative system

## Abstract

Climate changes cause various types of abiotic stress in plants, thus affecting plant growth and causing decline in yield. An urgent need exists to develop an environmentally friendly attitude based on principles of sustainable agriculture. Nanomaterials may improve plant growth and enhance crop productivity by handling the conditions considered stressful for plants in a sustainable and ecofriendly manner. Selenium (Se) has been put into the category of beneficial elements in plants. Se-enriched crops present a successful choice of dietary resource for Se-supplemented food and feed owing to their high bioavailability and accessibility. Researchers from distinct areas, including both nanoscience and plant science, should encourage emerging innovations that are linked with abiotic stress in crop production. The implementation of Se nanoparticles (SeNPs) is considered one of the predominating mechanisms by plants to ameliorate stressful conditions. Increasing evidence of earlier research revealed that SeNPs could enhance plant growth and development, nutrient bioavailability, soil fertility, and stress response while maintaining environmental safety. Meanwhile, some earlier studies reported that SeNPs might have a multilateral influence on plants dependent on diverse Se nanomaterial traits, doses, and plant species. More efforts are required to enhance the knowledge of how SeNPs impact crops exposed to different abiotic detrimental factors. In light of contemporary research challenges linked to SeNPs and the prolonged application of Se nanomaterials to plants, the aim of this review is elucidating the principal fruitful areas of SeNP exploration, comparisons with bulk Se, insights into mechanisms of abiotic stress alleviation in plants, existing research uncertainties, and practical challenges for SeNP applications under varying environments.

## 1. Introduction

Crop productivity contributes enormously to global food security and support of human wellbeing. Chemical fertilizers, pesticides, and other agrochemicals have a negative impact on ecosystems by reducing biodiversity, soil fertility, and by developing resistance against pathogens [1]. Among innovative agricultural technologies, nanotechnology has attracted more interest in the agricultural sector [2]. The term “nanotechnology” was coined by Norio Taniguchi [3]. Nanotechnology deals with manufacturing, research and manipulation of matter at nanoscale [4,5]. New prospects for integrating nanotechnologies in different fields have revolutionized the agroindustry. Agroapplicable nanomaterials could be used without altering the beneficial properties of the soil and could likewise protect plants from adverse environmental impacts; accordingly, nanoproducts preserve the health of the plant [6]. At present, selenium (Se) is referred to as a beneficial element in plants [7]. There is a necessity to develop deeper insight into the status of Se nanoparticles (SeNPs) in plant research (Figure 1), to make comparisons with bulk Se, and to identify the key knowledge gaps. Steps should be taken to accumulate and generalize previous research, and to update information on SeNPs’ interaction with plants under abiotic stress conditions. There is also a requirement for detailed data concerning ecological safety of the Se-containing formulations, including their implementation in cultivated plants, to elucidate consequences of chemically and biologically fabricated Se-containing product usage. The present review is aimed at gaining a better understanding of how nanoselenium benefits plants, at providing information on the proposed stress-alleviating mechanisms in plants exposed to the diversely originating Se agents. Generally, the use of commercially available nanoscale agrochemicals, nanoforms of plant fertilizers, fungicides, and problems of the plant disease management will not be considered.

## 2. Recent Reviews Dealt with Abiotic Stress in Plants Exposed to NPs: Novelty Statement

The current and future prospects of nanostructured materials in agriculture have been discussed in detail in notable review papers during recent years. The authors reported the opportunities and focuses of nanotechnological research purposed to improve a sustainable agricultural development (Table 1).

Kamran et al. [8] in their review discuss many chemical compounds of bulk Se available to plants, such as selenate, selenite, thioselenate, selenide, and elemental Se. This review article considers Se-mediated salinity tolerance and different strategies attained by plants under salinity. Nevertheless, this excellent review is aimed to exclusively elucidate the role of bulk selenium in salinity-exposed plants. Salama et al. [9] has discussed applications of nanotechnology in vegetable crops. The authors consider multiwalled carbon nanotubes, chitosan NPs (Ch-NPs), and NPs based on metals (manganese, molybdenum, silver) and oxides (copper, iron(II), silica, titanium(II), zinc). The content fits the keywords “nano-fertilizer, nanotechnology, productivity” being not specifically aimed at presenting data on SeNPs. The work by Rajput et al. [11] reports on the beneficial traits of the following NPs based on Si, SiO_2_, B, ZnO, Fe, Fe_2_O_3_, Fe_3_O_4_, Cu, CuO, TiO_2_, Co_3_O_4_, NiO, Al_2_O_3_, Ag, Au, Ce, as well as nanoparticulate materials based on carbon, zeolite, silicate, and chitosan in plants exposed to distinct abiotic stress. Three selenium-linked publications has been mentioned from viewpoint of cooperated action of the mixed SeNPs + SiNPs [10], Se/SiO_2_-NPs [12], Se- and CuNPs [13].

El-Saadony et al. in the comprehensive review [18] have highlighted the effect and scope of silicon-, Ag-, TiO_2_-, Zn-based NPs. Being structured on a basis of different stress conditions implementing distinct NPs, this review does not pay special attention to SeNPs. Recent advances in the nanotechnology application for abiotic stress alleviation have been elucidated by Javed et al. [19]. The review summarizes the potential role of NPs in plant abiotic stress resistance. To discuss a relevant role played by selenium, two research by Haghighi et al. [15] and Hernández-Hernández et al. [13] have been implemented. The review by Khalid & Iqbal Khan [21] considers three articles operating with SeNPs are cited, namely [20,22,23]. Stress of drought in plant impacted by NPs is reviewed by Kandhol et al. [24]. This review highlights the physiological, biochemical, and molecular mechanisms of NP-induced tolerance against water scarcity. Three cited Se-implementing works by Zahedi et al. and Ikram et al. are [12,25,26]. The review by Liu et al. [35] allows to improve our knowledge about bulk selenium impact on plants, while considers SeNPs in three works [34,36,37].

R. Periakaruppan et al. discusses many compounds containing metal or oxide, such as Cu, Zn, Ag, and TiO_2_ available as nanofungicides [2]. The review by Hayat et al. [38] regards the essence of abiotic stress as directly reducing the horticultural crop yield. The NPs employed in different plant species are metal oxides CuO, FeO, MgO, ZnO, Al_2_O_3_, non-metal oxide SiO_2_, as well as carbon nanotubes (CNTs), with no regard to selenium-containing ones. A comprehensive review by Shelke et al. [39] discusses secondary metabolite enhancement and environmental stress alleviation in crops using different mycogenic nanoparticles; however, it provides an exclusive overview of metal NP synthesis and the application of metallic NPs. Samynathan et al. [40] have presented a review of nano-Se relationship with plants. Phytonanotechnology uses plants and their components (roots, fruits, stems, seeds, and leaves) for the NP synthesis. The review article concentrates on phytofabricated SeNPs, their applications in plant growth, yield, and stress tolerance. The approach to SeNP synthesis using plant extracts is out the area of our review. Chandrashekar et al. [41] have presented a systematic review highlighting the beneficial role of metal or metal oxide NPs, and carbon-based nanomaterials. Outcomes of plant treatment by SeNPs refer to 4 research works, namely [20,25,26,42]. Al-Khayri et al. [43] consider nanoparticulate materials ZnO, TiO_2_, SiO_2_, Al_2_O_3_, hydroxyapatite, carbon, chitosan, and NPs based on Ag, Cu, Fe, Mn, Ce, K, Au, Co, and Ni applied under abiotic-stress-induced conditions in plants. The authors did not include SeNPs in the list of extensively researched NPs, and refer to 3 publications on selenium, namely [22,25,44].

A very recent review by Ain et al. [53] discusses how drought and salinity as co-existing stresses are detrimental to plants’ health and present an overview of the putative mechanisms underlying the NP-induced tolerance against these stresses. The authors discuss how ZnONPs, CuNPs, FeONPs, Fe_2_O_3_NPs, carbon nanomaterials SWCN and MWCN, CQD, silicon dioxide NPs, silver-containing NPs, MnNPs, and potassium-containing nanoforms could potentially enhance plant tolerance against drought and salt stresses; however, selenium-containing NPs are not included in this discussion. Excellent work by Yadav et al. [6] defines a distinctive term “plant nano-defense priming”, meaning that NPs prime plants for improved stress response, enhancing overall performance under hostile conditions. The review covers metallic, metal oxide, and carbon-based nanoparticles to enhance plant defenses. SeNPs are considered in relation to biotic stress by citing [54,55]. The biotic stress problem is untouched in context of our review. To the best of our knowledge, the review by Song et al. [56] presents a most comprehensive data concerned with SeNPs involvement in plant abiotic stress, being based on 18 research papers. This work summarizes research concerning feasible SeNP implementation in crop production.

As the Table 1 demonstrates, several meaningful contributions have been published related to distinct NPs—plant (including the abiotically stressed plant) interactions. Nevertheless, a comprehensive review is still needed regarding selenium-containing nanostructures, which mediate healthier development in plants. The present work follows a necessity to elucidate the role of Se, bulk form, compared to SeNPs, in the refinement of the growth parameters and biochemical characters of abiotically stressed plants, with the intention of identifying opportunities to conduct further research.

## 3. Effects of Exogenous Bulk Selenium on Plant Growth and Redox Characters

Plants adjust their vital approaches to survive under different abiotic stress circumstances induced by climatic variations, including drought, flooded or waterlogged conditions, salinity, uncomfortably low or high temperature, and the threat of toxic heavy metals [57]. For prolonged periods of time, a lot of studies have been conducted to unveil the biochemical consequences of the above stressful conditions, especially in endeavors to lessen losses in plant yield and quality by implementing the agents with phytohormonal, osmoprotective, antioxidative, and microelemental nature [8]. Trace element Se supplementation at a lower specific concentration is capable of protecting a plant from the oxidative deterioration induced by reactive oxygen species (ROS) by means of activating the antioxidative mechanisms, but also improves the Se level of dietary crops. Selenium in a plant-soil system further exerts the effect on humans and animals as higher organisms in the food chain via the Se content in consumed plant components [58].

Selenium is an essential micronutrient entering the composition of powerful natural antioxidants. Se was identified to enter the molecular structure of glutathione peroxidase (GSH-POx) about a half of century ago [59]. The problem with the organic Se fortification of functional tea and food could be potentially resolved by implementing the highly Se-tolerant, and thus Se-enriched, edible plants. There exist diverse pathways of Se usage in agriculture, those being related to Se applied to soil, seed treatment, culturing in Se-containing nutrient solution, and foliar spraying [60]. Biological activity of Se depends on its chemical form [61]. It is commonly accepted that inorganic Se substances and inorganic salts, with the most frequently used being sodium selenite (comprising Se(IV)) and sodium selenate (comprising Se(VI)), exert toxic action, whereas organic Se substances, in particular seleno-amino acids, have lesser effects, with a higher tolerance in plants. Thus, the Se-tolerant edible plants are applied to fortification and transformation of the inorganic Se compounds into organic ones with the preserved Se level in the functional food products. Selenium in its turn can enhance accumulation of some active phytocompounds.

Even though Se is considered a non-essential (only beneficial) element in plants, some data are indicative of Se’s useful impact. At lower concentrations, Se positively acts as an antioxidant, can stimulate the plant growth, inhibits lipid peroxidation and injury of cell membrane. While at relatively high contents, Se functions in a manner resembling a prooxidative behavior accompanied by negative consequences as for plant reduced yield. Chlorophylls and related pigments, alongside exhibiting a fairly appreciable Se dependency, are all recognized to possess different precious medicinal properties, including antioxidation, anticancer and antiaging effects. Contents of chlorogenic acid, carotenoid pigments and chlorophylls in the leaves of Se-fortified *Lycium chinense* L. enlarged to a great extent up to 200 or even 400 percent to the untreated control, thus being indicative of the given plant’s greater reserves and higher medicinal value than previously thought [61]. Additionally, a significant symbatic correlation was found between the value of ratio “chlorophyll a/chlorophyll b” and Se, between carotenoid content and Se [62], when the Se fortification caused the 200–300% increment in concentrations of these phytocompounds. Selenium was revealed to intensify respiration via stimulating an electron flow through a respiratory chain, that further promotes the biosynthesis of chlorophyll pigments. Meanwhile, sodium selenite was shown to enhance a *Brassica* plant respiration, whereas without an observable alteration in chlorophylls [63]. In this same bulk form, Se concentration not higher than 50 mg/L applied to soybean impacted positively a plant yield, influencing through a chlorophyll deterioration avoidance [59]. The correlation between selenite and chlorophyll biosynthesis needs further research.

Bulk Se has been found to protect lipids in plants from excessive peroxidation by means of both enzyme-mediated and non-enzymatic processes. The antioxidative effect of bulk selenium was for a long time recognized to be attributed to elevated superoxide dismutase (SOD) and GSH-POx enzymatic activities. The foliar application of Se (50 g per hectare) improved the antioxidant state characters (SOD, catalase, ascorbate peroxidase and glutathione reductase) in cowpea plants, and, along with mediating reactions catalyzed by these enzymes, led to a lowered concentration of malondialdehyde (MDA) and hydrogen peroxide [64]. Assisted by the Se antioxidative properties, the yield of several crops such as rice [65], cowpea [64,66], canola [67], wheat [68], and potatoes [69] were increased by soil or foliar Se application.

Therewith the essential improvement for the enzymatic component of antioxidants in plants correlated positively with Se level. Thus, selenate-exposed plants exhibited much slower decrease in antioxidative enzymes activity compared to control plants even several months after sowing [70], and under the senescence-induced oxidative burst [71]. The effect of sodium selenate biofortification on biomass yield, Se level, essential oils, and phenolic compound content, as well as on the antioxidant properties of basil (*Ocimum basilicum* L.) leaves was investigated [72]. Se application in the form of either nutritive solution or foliar spray at a level of 2 µm or 5 µm, respectively, in basil leaves appreciably increased Se concentration, the essential oils biosynthesis, hydroxycinnamic acid derivatives along with other phenolic compound production related to antioxidant activity as a whole. Thus, the works implementing bulk selenium clearly indicate the antioxidative role of Se. Its phytostimulating effect has been demonstrated by enhanced antioxidant enzyme production, diminished lipid peroxidation, lower hydrogen peroxide and other ROS concentration, and greater content of different chlorophyll-related pigments exceeding untreated plants.

One of the Se-linked research gaps to be filled consists in that only a few studies devoted to the improvement of fruit nutritive characters caused by any form of Se application have been performed, accordingly paying poor attention to biomechanisms behind Se action. Selenium pretreatments of crops were found to result in enrichment with ascorbic acid [73], soluble biopolymers as saccharides and proteins [74,75] and with Se as such [73,76]. Liang et al. [77] revealed the Se usage capability of significantly enlarging the leaf-and-stem level of ascorbic acid while diminishing the soluble protein and soluble saccharides level in the stem and leaf, respectively, in flowering Chinese cabbage. Foliar pretreatment with selenate increased Se content, as well as the content of amino acids, some monosaccharides as glucose and fructose, and of other substances as phenolic acids, kaempferol, quercetin, quercetin–hexose–deoxyhexose pentose levels [78], total flavonoid content, glutathione, vitamin C, and vitamin E in tomato (*Solanum lycopersicon* L.) fruit [73]. The application of bulk Se or sucrose, or concomitant use of both chemicals, exerted significant impact on the growth parameters and nutritive properties of pea (*Pisum sativum* L.) sprouts [79]. Comparison to Se individual usage showed that the appropriate external application of selenium + sucrose binary agent at 1.25 mg/L and 10 mg/L concentration, respectively, appeared to be more potent in improving nutritive traits than the above individual components. In lettuce, the value of overall soluble sugar parameter became greater under a foliar application of Se-containing solution; nevertheless, only a relatively low Se concentration resulted in higher levels of dietary fibers and ascorbic acid [80].

The research findings show that the impacts of diverse levels of inorganic Se-containing salts on the nutritive properties of cultured plants are distinct. At the same time, there are studies demonstrating that SeNPs provide more positive physiological effects in plants compared to Se inorganic salts.

## 4. Evidenced Transfer to Nanoparticulate Selenium in Favor of Plants

NPs exhibit several properties distinctive from those of conventional bulk materials owing to their nanosize [21]. Possible toxicity of some specific Se-containing chemical compositions must be taken into account in harnessing certain Se formulations. An alternative approach to aid in undermining the potential hazardous traits of Se is nanotechnology [81]. Fundamental knowledge in the discussed area is contributed by the observations derived from research carried out with the same plant object but different forms of acting material, containing or not NPs. Only a few researchers compared sodium selenate and Se-containing nanoforms’ effects with respect to plant development. Suitable concentrations of SeNPs or sodium selenate, being applied to seed priming under drought stress, increased germination percentage, and promoted physiological processes resulted in improved chlorophyll-related pigments, proline, and the principal antioxidative enzyme (catalase, ascorbate peroxidase, SOD) level in quinoa seedlings [42]. This study provides evidence that seed priming with SeNPs results in a positive effect exceeding sodium selenate usage.

Coffee supplementation with Se serves as a perspective approach in consideration of coffee beans’ high demand. Works dealt with the Se foliar treatment impact on coffee plants were absent in literature before the work by Mateus et al. [81]. This paper identified a research gap concerning Se supplementation consequences as for coffee yield and physiological characters of coffee plants. For the first time the study [81] has shown that foliar Se application using nano-Se and sodium selenate at a concentration ranged between 20 and 120 mg/L resulted in higher photosynthetic pigment content. This study also indicated that nano-Se (160 mg/L) liquids sprayed on coffee leaves promoted enhanced productivity.

Selenium as nanoforms displayed a remarkable ability to be dispersed, and to serve as antibacterials at a low concentration, much less environmentally toxic compared to inorganic Se salts [82]. It was found [33] that the treatment with Na_2_SeO_4_ or SeNPs led to increase in maturity index and mitigation of shattering pomegranate fruit, herewith the efficacy of the Se nanoform being higher than Se salt. Siddiqui et al. in the research concerned with the effects of both SeNPs and selenous acid on the barley (*Hordeum vulgare* L.) seed germination, roots and shoots growth, has elucidated a positive SeNPs impact on all these parameters. Moreover, it can be concluded that the nanoparticulate form of Se is less toxic compared to the selenous acid form, for ensuring intensified growth and better development of crops [30]. During tomato (*Lycopersicum esculentum* Mill. cv. ‘Halil’) vegetative growth, the increased chlorophyll content, enhanced plant growth and relative water content were detected as being caused by Se treatment. Nano-Se material was commercially available and additionally purified, the purity percentage consisted 98%, without the shell-forming, capping or stabilizing agent. However, even in this poorly bioavailable (compared to Se bionanocomposites [83]) form, selenate and nano-Se amended physiological responses in tomato plants caused by distinctly designed stressful conditions (exposure to pulsed or prolonged, elevated or reduced temperature) [15].

Efficient processing with exogenous Se aimed at mitigating a pesticide stress lacks systemic research till now, whereas Se has been proved to be able to alleviate an oxidative damage in the pesticide-affected crops. First research on the relationship between nano-Se (SeNPs, average size 50–78 nm) and tea quality under the prolonged oxidative stressful conditions caused by pesticide was attempted by Li et al. [84]. To this end, the consequences of the long-term application of imidacloprid, acetamiprid, and difenoconazole were observed. Nano-Se increased the nutrients and secondary metabolites of tea [84]. There are earlier findings that the overall amino acids and ascorbic acid content are elevated in green tea, whereas polyphenols level is substantially lowered by tea supplementation with the inorganic Se salts, selenites, or selenates. The authors stated that the outcomes demonstrated different effects on the regulation of nutrient components in tea, severely depending on the Se chemical forms and contents, first of all SeNPs and Se(IV), Se(VI) salts. Such enrichment of products in nano-Se should be regarded as a kind of dietary fortified food, especially for patients with Se deficiency [84].

In general, nano-Se is more active than bulk selenium, moreover, the SeNPs produced by microorganisms contribute to ecological safety. El-Saadony et al. [20] employed biological SeNPs (BioSeNPs) synthesized by *Lactobacillus acidophilus* ML14, which was the pioneering available study concerned with plant disease management by means of the application of SeNPs fabricated in the course of a bacterially mediated procedure. The application of SeNPs produced with the aid of metabolites of *Lactobacillus casei* enhanced *Chrysanthemum* tolerance to elevated temperature (at 40 °C) for different *C. morifolium* Ramat varieties, being obviously mediated by the antioxidative state improvement, photosynthetic processes promotion, and increase in the total chlorophyll content [34].

The mushroom-assisted green route has led to Se bionanocomposites comprising red Se(0), which could be successfully implemented in plant growth promotion [85], largely thanks to their biocompatible mycosynthesized stabilizing matrix [86]. Various ways of synthesizing SeNPs have been discussed; among them, a green strategy based on nano-Se biomanufacturing using fungi is thought to be very attractive. This mycosynthetic process demonstrates intrinsically reduced ecological issues and enhanced biological activities [40]. Particulate selenium supplementation is highly applicable as an approach that increases plant stress tolerance and positively influences plant metabolism (Figure 2). SeNPs can promote growth, enhance plant resistance to oxidative stress by adjusting antioxidants level, improve soil nutrient status, and participate in the transpiration process. In general, a holistic positive influence on plants is often a distinguishing feature of nanoformulation compared to the bulk material.

## 5. Se Nanomaterial Participation in Mechanisms Providing Plant Tolerance to Abiotic Stress

### 5.1. Overall Mechanism Behind Nanomaterial-Sustained Mitigation of Abiotic Stress

Plants experience a broad set of adverse abiotically mediated stress factors, such as drought, salinity, waterlogging, flooded environments, uncomfortably decreased or elevated temperature, harm of toxic heavy metals, and ultraviolet radiation. Generally, abiotic stress exerts a negative influence on sustainable plant production and development in agriculture, thus affecting food security, and leading to economic losses [18]. SeNPs induce stress-mitigating distinct morphological, physiological, and biochemical improvements in plants.

Drought stress heavily impairs cellular and metabolic functions, ionic exchange, membrane continuity, and structures of plant biopolymers. That is followed by the reduced plant growth, attenuated photosynthesis, and decreased enzymatic activities. Additionally, drought-enabled more active production of ROS in plants leads to heavier oxidation stress [41]. Different NPs are capable of regulating both the drought-responsive gene expression and different phytohormones biosynthesis, eventually helping plants to withstand water deficit stress. To mitigate the osmotic stress caused by drought conditions, NPs switch on the diverse mechanisms resulted in the enhanced root propagation, aquaporins upregulation, intracellular water uptake improvement, compatible solutes biosynthesis, and ion homeostasis maintenance [24].

Salinity is one of the key deteriorating factors interfering in plant growth, yield, and physiology [87]. Salt stress impairs seed germination, plant development, and crop productivity. The world situation with regard to damages caused by salinity refers to the whole set of examples. Thus, not less than 20% of agricultural soils and about 33% of irrigated farming areas are unsuitable for cultivation because of high salinization. Shifts in ionic equilibria arouse enhanced oxidation and different alterations in the vital processes. Furthermore, sodium cations substitute other positively charged ions (primarily potassium cations) in biochemical reactions, followed by impending Na+/K+ ratio displacement toward sodium ions excess, and by the metabolic processes affected.

Metabolically mediated Se nanoformulations’ effect on plants contributes greatly to the ability to withstand abiotic stress in plants. Thus, Li et al. [84] reported that the foliar application of nanostructured Se enabled the improved tea tolerance in response to pesticide stress. Therewith the proposed biochemical action of nano-Se comprised the Se-induced considerably enhanced overall content of tea phenols and flavonoids, that assisted in coping with the stress-factor-induced excessive oxidation state. Phenolics-sourced and others secondary metabolites participate as a component in the anti-stress defense system in plants [88]. Underlying pathways by which the nano-Se interventions are implemented in the secondary metabolite production in plants still remain unveiled [41]. Mycogenic NPs elicit plant tolerance towards abiotic stress. Utilizing mycogenic NPs enhances secondary metabolite production. NPs play role of binding and transporting agents with respect to nutritional substances and plant growth effectors, whose uptake and targeted transfer occurred to be facilitated by these means [39]. Selenium nanoparticulate materials boost biosynthesis of the specialized metabolic compounds that are frequently essential for plants in their defense against stressful environments. The said compounds could be of phenolic, flavonoid, carotenoid and terpenoid chemical nature, to which the improved antioxidative traits and nutritive value are often attributed in current literature.

Utilization of SeNPs enhances the antioxidant abilities of treated plants owing to Se involvement in cellular redox reactions. For instance, SeNP application to tomato leaves resulted in the considerable activation of enzymatic redox pool and the increase in phenolics and amino acids level in plants [13]. A significant increase in alkaloid contents is also reported in plants subjected to stress [41]. Alkaloids were retained in higher abundance in roots compared to shoots. Greater alkaloid accumulation positively correlated to improved osmotic adjustment, thus leading to enhanced water uptake and better plant state. Critical need for continued exploration of the Se nanostructured materials’ impact on mitigating stressful experiences in plants remains for insight into the mechanisms behind these effects. It is currently clear that applying SeNPs to plants for better withstanding to abiotic stress provides distinct advantages in terms of plants’ physio-morphological characters, proper nutritive supply and photosynthesis process, osmolyte concentration, non-enzymatic antioxidants level and antioxidant enzymes activity.

### 5.2. Se Nanomaterials in Promoting Plant Growth and Yield

Enhancing plant growth and productivity is one of the principal lines for nanotechnology in agriculture. To this end, different approaches to the NPs introduction in plant could be taken into consideration to challenge the interference with stressful environments [11]. Application of chemical and biogenic nanoscale materials could be an efficient adaptation strategy to manage and endure adverse abiotic factors, thus improving plant growth indices, enhancing biomass, and achieving sustainable production [89]. Dietary supplementation by selenium as a chemical element essential for humans and animals could be provided by the application of SeNPs serving as an additive during plants cultivation, followed by alleviating the nutrient deficiency symptoms in a subsequent trophic chain.

As has been found by several studies on plant growth and yield, SeNP additions improved the quality and productivity of many crops, such as potato, pumpkin (*Cucurbita pepo*), canola (*Brassica napus* L.), soybean, and cluster bean (*Cyamopsis tetragonoloba* L.), at a low concentration [59,63,90,91]. Furthermore, Hernández-Hernández et al. [13] showed a 21 percent increment in tomato (*Solanum lycopersicum* L.) fruit yield obtained by using the SeNP treatment at a 10 mg/L concentration. This nanomaterial was capable of balancing the chlorophyll-related pigments, ascorbic acid and GSH levels, as well as of regulating the antioxidative enzyme activity, thereby exerting a bilateral protective effect on the side of low-molecular-weight and enzymatic components of the antioxidant defense system in plant.

Ikram et al. [26] reported on the positive impact of the SeNPs’ foliar applications on distinct developmental and productivity-related parameters of wheat (*Triticum aestivum* L.) plants under drought stress. Se nanomaterial was biofabricated using water extraction of *Allium sativum* L. buds. The resulting extract served for selenium-ions chemical reduction followed by NPs stabilization assisting in the process of SeNPs formation. The outcomes of the foliar application of the thus phytoproduced SeNPs to wheat seedlings subjected to drought stress testified to the beneficial changes in the seedlings’ physiomorphological parameters. Thirty mg/L of biosynthesized SeNPs exposure during treatment appeared to be the most efficient and resulted in the improved morphometric indices of wheat plants, as compared with other tested Se amounts under water deficit conditions.

Under saline conditions, SeNPs were beneficial with respect to more active growth, improved ionic selection in roots, and increased productivity of rice [28]. Report by Boroumand et al. [92] concerns a green synthesis of SeNPs through chemical reduction method using selenite as a reactant to be reduced, ascorbic acid as a reluctant and polyvinyl alcohol or chitosan as stabilizers. For comparison, SeNPs biomanufactured by lactic acid bacteria (Bio-SeNPs) were prepared. Bio-SeNPs appreciably enlarged spike, shoot, root length and biomass value, number of grains per spike, and 1000-grain mass at a SeNP concentration of 100 mg/L, with an increment near 20% exceeding the SeNPs synthesized chemically and applied at the same 100 mg/L-level. Under the stressful conditions accompanied by the enhanced ROS generation, SeNPs appear to be an effective stimulant for the plant production and yield. The treatment of barley (*Hordeum vulgare* L.) seeds with SeNPs resulted in better development and morphological traits, therewith the barley seeds’ germination capacity and energy were improved in seeds processed with 5 mg/L-SeNPs solutions [47].

Thus, SeNP application improves growth traits under various stresses [38]. Current research denotes a positive influence on crops with respect to seed germinability, growth, and other agriculturally relevant indicators shown to be associated with Se-containing nanomaterial treatment. The above effects are mediated by the SeNPs’ interaction with plant cells [41], thereby promotion of water and nutrients uptake and improved development in plants.

### 5.3. Se Nanomaterials in Regulation of Nutrient Imbalance, Cell Integrity, and ROS Homeostasis in Plants

The contemporary demand for nutrient-supplying approaches implies the urgent necessity to invite novel findings on how to mitigate negative impact of nutrient leaching to outdoor, for addressing the challenge of these loss preventions [9]. To this end, the nutritive substances in their nanoparticulate form should be taken into consideration as the nano-agents capable of releasing slower for prolonged time compared to ordinary nutrients. For that manner the diminished losses from the soil positively correlate to decreased contaminants in the surrounding environment [93]. Nanostructured type of a large number of plant nutrients could serve as a beneficial form for improved foliar delivery owing to its transport-linked intrinsic spatial–temporal properties [17]. An important trait of nanomaterials is their capability of aiding in water retention by modifying the plant cell wall properties and regulating the stomata opening, that results in greater water usage efficiency [41]. Selenium as a component of NPs could be involved in transferring nutritive material and absorbing nutritive molecules within the parts of plants [21], eventually perfecting the agri-important attributes [94]. Among different consequences of abiotic stress, one at the top of the list is nutrient imbalance, which contributes to the food security-threatening crop losses throughout the world [21]. SeNPs were found to exert a positive impact on the stress-affected plant state, not only by providing a more facile uptake and transfer of essential nutritive agents, water, and minerals, but also with the additional effect of inhibiting lipid peroxidation [40]. Foliar application of SeNPs (20 mg/L) to pomegranate (*Punica granatum* L.) resulted in the increased level of phenolic compounds and chlorophyll-related pigments, adjusted osmolyte content and nutrient status compared to those characteristics in untreated trees [25]. SeNPs reduced the water-deficit-initiated stress-responsive increase in hydrogen peroxide content and lipid peroxidation extent through the improvement in antioxidative state, thus substantially contributing to cell stability enhancement. Salinity conditions exert negative effects on plants in various aspects, including negative changes in carbohydrate and protein metabolism. Se supplementation was proven to be useful in managing the essential macroelement retrieval from soil, followed by the enhanced metabolites biosynthesis, and better stress signaling functioning to induce salinity tolerance in wheat (*Triticum aestivum* L.) [95]. A solution containing 10 mg/L of SeNPs sprayed on the leaves of *Coriandrum sativum* was capable of increasing the plant’s soluble sugar content by 1.5 times [36].

Several research consider the Se action as showing a direct relevance to the antioxidative system regulation and cell structure maintenance [73]. Temperature elevated above a critical threshold for a prolonged period enough for permanently providing insufficient plant growth conditions is often referred to as heat stress [96]. The application of SeNPs contributes to heat stress tolerance in plants. Membrane damage, poorer pollen germination and crop yield in sorghum plants affected by heat stress was found to be recovered by the application of SeNPs via activation of the antioxidant defense system [22]. When added to sorghum crops, SeNPs reduce ROS production caused by elevated temperature to enhance antioxidant defense.

Cells’ antioxidant responses, established with the intention of equilibrating yield, detoxifying ROS, and reinstating pro-/antioxidant processes homoeostasis, generally utilize enzymes such as SOD, GSH-POx, catalase, peroxiredoxins, and other peroxidases. Additionally, the ascorbate–GSH cycle and its enzymatic components are implemented [43]. Additionally, the non-enzymatic compounds such as GSH, vitamin C, carotenoids, tocopherols, and phenolics participate in the plant antioxidative system. SeNPs are found to be the inducers and complementing players mediating this antioxidative response owing to Se’s dual properties as an ROS generator and scavenger, as well as its ability to regulate membrane damage [97]. The latter seems to be a consequence of susceptibility dependent on the composition of membrane lipidic pool and on the cellular organelle ultrastructure and its association with redox system [22]. Presumably, SeNPs’ interaction with plant boosts a whole range of responsive reactions at a cellular level, including those from a defense system also managed by stress biomarker synthesis. Ultimately, this induces the production of both non-enzyme antioxidants and antioxidative enzymes. The enhanced activity of the enzymatic pool of the redox system positively influences the stress tolerance relying upon the alleviated oxidative deterioration of membranes. SeNP supplementation mitigates the pesticide-caused oxidative damage, allowing plants to achieve sustained cellular integrity by reducing the hydrogen peroxide, superoxide radical anion, and MDA contents, while increasing the GSH-POx, peroxidase, SOD, and catalase levels in tea leaves [84].

Recently, some reports have stated that the application of nanomaterials including SeNPs directly influences the antioxidant enzymes activities under a stress environment [19]. ROS, both of the non-radical (as hydrogen peroxide) and radical nature (superoxide radical anion and hydroxyl radical), are toxic byproducts of cellular oxygen metabolism. The intracellular ROS level is thought to be a determining factor inducing apoptotic state, senescence, and cell cycle arrest. Selenium has a remarkable role to play in many aspects of enzyme-catalyzed reactions mediated by POx, SOD, catalase, ascorbate peroxidase, and GSH-POx, and of non-enzymatic processes mediated by low-molecular-weight antioxidants, all of them helping to combat the abiotic stress-induced overproduction of ROS. The latter are responsible for agitating plant cell integrity [8]. The sustained homeostasis of ROS and of the photosynthetic functioning is among the principal priorities of plants subjected to abiotic stress [98].

### 5.4. Se Nanomaterials in Improving Photosynthetic Efficiency in Stressed Plants

Abiotic stress consequences decrease the photosynthesis efficiency, life cycle duration, and productivity of crops. SeNPs as environmentally friendly agents are capable of mitigating several stress types. High-temperature stress (heat stress) decreases photosystem II quantum yield and photosynthesis [22]. Drought stress is a significant concern disrupting the functions of photosynthetic activities. To resist water stress, plants reportedly decrease a stomatal conductance to manage the amount of vaporized moisture, and the volume of carbon dioxide assimilated that enable escape of photosynthesis inhibition. By means of increasing the photosynthetic activity, NPs could ameliorate drought-induced reduction in carbon assimilation [24]. Salinity is an extremely destructive type of abiotic stress seriously threating plants with respect to their morphophysiological and metabolic characters, therewith the photosynthetic system being collapsed, the photosynthesizing area of the leaves reduced [57]. Salinity conditions negatively interfere with the growth rate, water absorption capacity of plants, leading to senescence [23]. Additionally, salt stress influences the metabolic and physiological functions by increasing levels of sodium cations and chlorine anions, that inhibit growth and photosynthesis [99].

Selenium foliar application was capable of supporting optimal level of photosynthetic pigments in several cultured plants, such as wheat [95,100], cowpea [64,66], canola [67], tomato [101] and lettuce [102]. Researchers [103] revealed that SeNPs supplied to peanut helped with increasing in a chlorophyll level. That was attributed to a positive Se impact on the relevant chloroplast enzymes implemented in the chlorophyll biosynthesis. The resulting higher concentration of carotenoids and chlorophyll b was indicative of a stronger photoprotective action of selenium on the antenna complex [66,104]. The plant species *Cyamopsis tetragonoloba* L. Taub, cluster bean, was studied by Ragavan et al., 2017 [91] during its 60-days growth. Growth, biochemical characteristics and yield of cluster bean were explored after treatment with SeNPs synthesized using a precipitation method by sodium selenite reaction with ascorbic acid. In the 60-day plants treated with one and the same SeNP concentration, the levels of chlorophyll-related pigments and carotenoids, anthocyanin-group compounds, protein, unbound amino acids including L-proline, and nitrate content of leaves exceeded those in the control (untreated plants). Application of different nanoparticles including SeNPs corresponded to the elevated chlorophyll pigments and lycopene concentration, as well as to the upregulated activity of GSH-POx enzyme in the bell pepper leaves [23]. Nanostructured Se benefited the physiological system altogether and a development of Chinese cabbage [105].

In spite of repeated confirmations that SeNPs advance the photosynthetic capability and antioxidative status, the mechanisms underlying this nanomaterial impact on the photosynthetic apparatus and pro-/antioxidant balance yet stay unexplored. It was found that the weakened photosynthesis ability in wheat affected by the elevated temperature conditions should result from the lipids’ modifications towards higher proportion of unsaturated bonds in molecules, thus facilitating their oxidation, acylation, and organelle injury [106]. Heat stressful environment deactivates to a significant extent the antioxidative enzymatic pool including SOD, catalase, and POx. High-temperature stress alleviation by SeNP treatment in grain sorghum has been reported to be mediated by the increase in phytohormone cytokinin playing a prerequisite role in repairing the photosynthetic pigments under stress [22]. Hernández-Hernández et al. reported that SeNP application restored the chlorophyll level in tomato leaves [13]. According to Elkelish [95], wheat seedlings exposed to 5 µm-selenium-containing solution exhibited the improved capacity to bioproduce overall pigments, specifically carotenoids with the percentage increment being 13% and 8%, respectively. Furthermore, the photosynthetic rate and stomatal conductance indices gain the values increased by 23% and 19%, respectively, versus control plants. SeNPs biofabricated by lactic acid bacteria increased the content of photosynthetic pigments [20]. The overall chlorophylls and carotenoid pigment content in wheat processed with BioSeNPs (100 mg/L) was 12% to 32% greater compared to the control. The authors additionally pointed out the regulated gaseous exchange characteristics (transpiration, net photosynthesis, and stomatal conductance), which were also superior to the control.

### 5.5. Se Nanomaterials in Regulating the Anti-Stress Compound Production and Hormone Balance

The anti-stress metabolites biosynthesis induces pathways associated with tolerance to abiotic stress in plants. Plants produce osmoprotectants and osmolytes (amino acids, primarily proline, soluble sugars and sugar alcohols, other polyols, and quatemary amines) to protect cellular membranes and proteins exposed to the denaturing environment. A sufficient build-up of osmolytes provides highly efficient sustaining for osmotic equilibrium and oxidation protection for primary metabolites in plant defense against oxidative deterioration, thereby alleviating cell damage [107].

Recent reports dealt with plants exposed to water-deficit stress demonstrated the enlarged content of free amino acids, specifically proline, overall proteins, and carbohydrates, at the NP treatment [108]. Proline is a critical osmoregulatory agent inducible by drought stressors. Nano-Se boosted the development and yield of cucumber plants co-stressed by excessive salt and elevated temperature [97]. The foliar treatment by SeNPs was revealed to decrease the damage effect of high temperature on cucumber, and Se treatment promoted the positive regulation of proline, POx, catalase activity, and relative moisture level. Treatment with SeNPs boosted beneficial changes in the low-molecular-weight metabolite pool of fruits, namely phenolics, flavonoids, carotenoids, and GSH [23]. Additionally, SeNPs enhanced the antioxidant enzymatic activity, that is essential for the detoxification of ROS and lipid peroxidation damage avoidance.

Proline is a molecular chaperone implemented in a protein integrity support mechanism as an osmolyte essential for sustained osmotic pressure, and SeNPs could increase proline content by regulating gene expression in proline biosynthesis. Consequently, the improved physio-morphological characteristics in crops could be provided by SeNP-mediated increase in proline level [109].

Plant hormones are among the key players in plant signal recognition and transduction. SeNP-primed seeds were shown to exhibit greater germination rates, obviously owing to more active production of germination promoters, namely phytohormones [42]. The hormones jasmonic acid, abscisic acid (ABA), and salicylic acid were found to manage distinct regulatory pathways assigned to stressful conditions [110,111], yielding alterations in nutritive components, metabolism, defensive systems and expression of stress-related genes. ABA was reported to be crucially valued phytohormone behind the mechanisms appointed in plants to withstand abiotic challenges, with no exception for drought. ABA alleviates water-deficit-induced stressful state, and assists in sustaining tolerant plant life, for which ABA controls root development, leaf elongation, and plant extension as a whole [112]. The study [113] indicated that the NP-processed drought stressed plants were characterized by improved tolerance owing to the selenium-upregulated gene expression of drought-responsive genes, and to the enhanced bioproduction of ABA. The latter also positively influenced stomatal conductance, thereby managed transpiration rate and cellular moisture retention [25]. In the environment stressed by severe drought, ABA meanwhile played the ascertained role of a signaling molecule involved in the production of other phytohormones exemplified by gibberellins, cytokinins, and the gaseous hormone ethylene [112].

Plants respond to drought stress conditions and activate defensive biochemical processes, for the most part under the influence of auxins and gibberellins in particular, confirming their efficiency in promoting plant tolerance to abiotic stress. Auxin, cytokinin, gibberellin, abscisic acid, and ethylene are several compounds of great importance that manage cell reactions vital for plant development [114]. A lot of research has confirmed the relevant auxins effect, especially of indolyl-3-acetic acid (IAA), which manages cellular functioning (division, expansion, and differentiation), being implemented in the principal metabolic activities to alleviate abiotic stress consequences in plants [115]. It was hypothesized that NPs could activate production of the essential amino acid tryptophan indispensable for the most potent auxin biosynthetic pattern to yield phytohormone IAA influencing various sides of physio-biochemical functions [116].

Nano-Se modulated the expression of ABA and gibberellic acid genes during the seed germination stage in rapeseed (*Brassica napus* L.), enhanced the germination attributes and seed microstructure, and reduced the oxidative damage under salt stress [87]. These authors validated again a beneficial effect of SeNPs on adjusting a gibberellic acid level, that promoted the early seedling growth for the nano-primed seeds. Gibberellic acid along with ABA participate in mediating defensive reactions to plant-threating abiotic stress; meanwhile, these acids’ equilibration is extremely important in the course of seed germination. ABA is thought to be a potent regulator of the abiotic stress response and a key inducer of the dormancy state of seeds. Selenium nanoformulation manifested a growth-promoting effect caused by the adjustment of jasmonic acid, salicylic acid, and ABA levels. Li et al. [117] found that the Se nanoform introduced to pepper plants at a concentration of 20 mg/L enhanced jasmonic acid content. The selenium-induced expression of this acid-responsive gene and the biosynthesis of the corresponding enzymes are critical for maintaining the oxylipin production in plants [118]. By means of altering the expression of genes involved in ABA and salicylic acid biosynthesis and upregulated with SeNPs, these acids’ levels were adjusted. Additionally, the wheat plants treated with SeNPs exhibited the upregulated expression of heat shock factor A4A, responsive to multiple abiotic stresses [119].

In adverse environments, the non-enzymatic component of plant antioxidative defense system appears to be harmed [120]. Phenylalanine ammonia-lyase (PAL) is the key enzyme of the phenylpropanoid synthetic route required for secondary metabolite biosynthesis, including low-molecular-weight antioxidant substances such as phenols and flavonoids maintaining the cell status of the stressed plants. The homeostasis between ROS and polyphenols reportedly provides the salinity-stressed plants recovery [121,122]. Total polyphenols as a component of plant redox system are regarded as both the scavenging ROS substances and the substrates of POx-like enzymes [123,124]; therefore, the stressed plant responses are dependent upon the phenolics biosynthesis induction. The application of Se/chitosan-based NPs significantly upregulated *PAL* expression and chlorophyll levels as assessed in stress-free and stressful salinity environments of bitter melon plants [49], and significantly raised the concentrations of relevant low-molecular-weight agents in bitter melon fruit [37]. Abedi et al. [125] reported that nano-Se upregulated *PAL* transcripts ascribed to secondary metabolism in chicory (*Cichorium intybus* L.). Selenium in its nanosized state also enhanced the expression of *PAL* in bitter melon plants [126]. Li et al. [117] concluded on the SeNP-caused improvements in the capsaicinoid biosynthetic path-related gene expression including *PAL*, and in the capsaicinoids and other metabolite production in peppers. SeNP-processed lemon balm (*Melissa officinalis* L.) was featured by the enhanced secondary metabolism via stimulating the rosmarinic-acid-synthase- and hydroxyphenylpyruvate-reductase-related gene expression [127]. Treatment with SeNPs was beneficial for *M. officinalis* with respect to the enhanced tolerance to salinity stress relied upon the activated antioxidative enzymes, primarily catalase, SOD, and POx, and upon the mitigated oxidative stress. Ghasemian et al. [128] reported that the SeNP-mediated increase in the antioxidant activity resulted in repaired lipid peroxidation and more salinity-tolerant *M. officinalis* plants. Furthermore, within an environment featuring excessive salt, SeNP additions were responsible for higher expression levels of rosmarinic acid synthase-responsive and PAL-responsive genes.

## 6. Synergistic Effect of Distinct Non-Metallic NP Application in Plants

Zahedi et al. [129] found that in strawberry (*Fragaria* × *ananassa* Duch.) plants, the foliar treatment using SeNPs suspension at 10 and 20 mg/L level enhanced the activity of antioxidant enzymes, foremost peroxidase and SOD, thus significantly lowering the stress-impacted lipid peroxidation indices (assessed by MDA), as well as a hydrogen peroxide content under both control and salinity stress conditions. In transferring to the cooperative action of SeNPs and silica nanoforms, with both combined to form the chemically manufactured Se/SiO_2_ NPs, Zahedi et al. [12] found that the foliar application of NPs under question recovered the developmental and productive characteristics in strawberry plants exposed to drought. According to this research outcomes, severe water-deficit conditions affected *Fragaria ananassa* with respect to chlorophyll concentration, which was nevertheless regulated as a consequence of the Se/SiO_2_ NP spraying procedure. These NPs’ implementation also led to the considerable positive alterations in such important redox-responsive characters as MDA, hydrogen peroxide, and enzymatic activities. Also observed were the maintenance of high osmolyte (carbohydrate and proline) concentration and elevated antioxidant enzymatic activity (SOD, catalase, ascorbate peroxidase, and GSH-POx). This in sum led to scavenged ROS, as a declined H_2_O_2_ content and lipid peroxidation indicated. Maximal benefits to plants exhibited by Se/SiO_2_ NP treatment at a 100 mg/L level were estimated via more preserved photosynthetic pigments compared to control plants, increased relative water content, and membrane stability index [12].

SeNPs, chitosan, and Se/chitosan NPs as the nanostructured conjugates were detected to be potent mediators in mitigating the adverse impact of abiotic stress on the cultured plants, capable of subsequently improving the developmental parameters. Se/chitosan NPs’ application could lead to better performance and health of plants suffering from salinity stress. A few years ago, Sheikhalipour et al. [37] showed that the Se/chitosan NPs’ application positively affected growth, yield, and antioxidative state, linked to both low-molecular-weight compounds and enzymes, in bitter melon (*Momordica charantia*) plants under salinity conditions, but also diminished H_2_O_2_ and MDA levels. In more recent study, Sheikhalipour et al. [49] conducted research focusing on the same plant bitter melon (*Momordica charantia* L.) seedlings and demonstrated that priming with Se/chitosan NPs could lead to the essentially improved characters attributable to growth and photosynthesis processes. Meanwhile, the above nanopriming strategy efficiently enhanced the antioxidative enzymatic traits including the activity of SOD, catalase, and ascorbate peroxidase, under the stressful salinity exposure applied to the seedlings. This research also observed the appreciable benefits in terms of the morphometric parameters such as shoot length and shoot fresh weight for the salinity-exposed seedlings. These crucial improvements might be owing to the reduction in oxidative-stress-linked attributes. Photosynthesis machinery components were inversely increased.

Negative consequences caused by a heavy-metal-affected environment present global concerns that challenge plant development. Abiotic stress in plants induced by heavy metal contaminants (HM stress) is a crucial issue worldwide [11]. HM stress causes oxidative processes, ROS accumulation causing significant harm to the plant morphology and functions [7]. The specialized properties of nanomaterials, particularly a sustained complexation with heavy metals and scavenging ROS, can be put into agricultural practice to alleviate HM stress in plants. Many researchers suggest the weakened toxicity of heavy metals following interaction with nanomaterials mediated by distinct mechanisms [130]. Plant production under different heavy metal stressful conditions could be considered a supportive approach implementing the chitosan-conjugated nanoform of Se.

Among heavy metals, Cd and Pb are commonly ranked among the most harmful contaminants, severely affecting both soil media and plant growth [10]. In *Coriandrum sativum* under cadmium stress conditions, the treatment with SeNPs (10 mg/L when calculated in Se content) enhanced catalase activity by 76% compared to untreated plant [36]. Li et al. studied the impact of SeNPs (1, 5, and 20 mg/L) under the heavy metal stress (Cd-contamination of soil) on pepper plant metabolism and pepper nutritional quality [52]. SeNPs at an applied concentration of 5 mg/L induced the increased contents of capsaicin and its derivatives by 29 to 45 percent depending on the compound. Besides, the composition of volatile organic compounds concerned with this crop quality and resilience, was significantly improved. Azimi et al. established that the Se/chitosan NPs modified photosynthetic potential, antioxidative enzymatic activity, and the essential oils pool in the Moldavian balm (*Dracocephalum moldavica* L.) subjected to Cd-caused heavy metal stress [50]. The efficiency of Se can be enhanced through its combined formulation with chitosan, thus alleviating the Cd-stress-triggered negative consequences. Satisfying progress on the way to solving a challenging problem linked with cadmium and lead toxicity in rice (*Oryza sativa* L.) emerged from foliage application of Se- and SiNPs. The authors check the efficacy of these nanoparticle types, single or concerted, on the HM content and the Se-enriched rice (*Oryza sativa* L.) yield. The research was carried out at the cadmium- and lead-contaminated paddy field. Se nanomaterial was chemically synthesized via the reaction between selenous acid with sodium thiosulfate, while SiNPs were commercially available. Single and combined NP treatments showed that in principle, the foliar application of SeNPs and SiNPs constitutes an efficient strategy to diminish the HM level in brown rice cultivated in Cd- and Pb-polluted soils. However, a concerted action of SeNPs and SiNPs appeared to be a key factor in reduced HM accumulation. The combined SeNP and SiNP implementation clearly outperformed a distributed design with respect to the rice quality and Se content. The synergistic effect of diverse non-metallic NP treatments in plants still needs further explorations implementing a holistic approach to make the practically oriented conclusions.

## 7. Se Nanomaterials’ Adverse Effects Involved in Plant Response to Abiotic Stress

Since the responsive reactions of each cultivar to SeNPs are severely distinct, that should be taken into account. “A spoon of tar spoiling a barrel of honey” can be met occasionally when considering SeNP application in plants. Under stressful conditions, the improvements in photosynthetic activity have been reported to be closely linked to the enhancement of the antioxidative-system-related metabolite and enzyme levels. Thus, Morales-Espinoza et al. [131] showed alterations in the content of chlorophylls a, b and (a + b) equal to 71%, 120%, and 97%, respectively, when SeNPs (20 mg/L) were applied in tomato plants exposed to salinity stress (caused by 50 mm NaCl). These researchers showed that when applying SeNPs to stressed plants, the level of antioxidants such as flavonoids, lycopene, and β-carotene, in tomato fruits was higher. Simultaneously, growing antioxidative enzymatic activity in tomato fruits took place, specifically of ascorbate peroxidase, catalase, and SOD, involved in ROS detoxification and in the membrane lipids peroxidation prevention.

However, the chlorophyll concentration dropped when the Se-treated plants did not experience excessive stress including salinity. That might be associated with the fact that SeNPs can initiate oxidative stress subject to these nanostructures’ features and plant species, accompanied by the induced chloroplast membrane peroxidation, and chlorophyll decay. Forced by the decreased chlorophyll concentration, plant carotenoids as multipurpose pigments operate as compulsory antioxidants to protect chlorophyll from further oxidative damage raised by such abiotic stress as drought [41]. It is known that the content of chlorophylls, as well as the level of accessory pigments (carotenoids, lycopene, β-carotene), in the leaves all play indispensable roles. Therewith the accessory pigments are powerful antioxidative substances acting through the excessive excitation energy dissipation in PSII, and so protect the photosynthetic system, specifically in the affected environment [23].

The same trend featuring the antioxidative potentiality of SeNPs in withstanding ROS while escaping DNA damage and cell senescence, was monitored by the application of Se-containing formulation to peanuts, accompanied by the lipids and proteins modified composition, and by the alleviated ROS-related stress [132]. The equilibrated ROS production and adequate antioxidative enzymatic activities play a decisive role in the cellular choice between redox signaling pathway and damage. Processing with SeNPs (20 mg/L) resulted in an increment of POx and ascorbate peroxidase activities, but accompanied by a decrease in catalase activity. The increase in POx and ascorbate peroxidase activity could presumably be the main prerequisite for hydrogen peroxide decay, the more so when catalase is inactive. Meanwhile, the application of 40 mg/L concentration of SeNPs resulted in a decline in both ascorbate peroxidase and catalase activities, probably due to a higher activity of POx enzyme, and to the mediatory function of Se in detoxifying pathways directed to prevent or to influence oxidative stress. Since SeNPs are known to regulate plant growth by modulating the signaling pathways in a ROS-dependent manner [132], the status of pro-/antioxidant activities balance is of importance. SeNPs induce alterations in chloroplast’s ROS level, that might be associated with changes in chloroplast functioning [133]. Hussein et al. [103] applied foliar treatment by SeNPs (0, 20, and 40 mg/L) during vegetative stage of three different groundnut (*Arachis hypogaea* L.) cultivars. SeNPs were chemically synthesized via the reaction of ascorbic acid with sodium selenite in solution. SeNPs -treated Gerogory cv. and Giza 6 cv. revealed better growth behavior compared to control plants; however, the morphometric indices of NC cv. exhibited a rather poor impact resulting from the SeNP implementation. The outcomes of SeNP application to groundnut cultivars with respect to plant development were attributable to physio-biochemical characteristics, namely modulation in photosynthetic apparatus, lipid peroxidation, antioxidant enzymes (catalase, POx and ascorbate peroxidase), pool of phenolics, flavonoids, and soluble saccharides. SeNPs as nanofertilizers, in principle, could mitigate an adverse impact of oversalted water on vegetable yield and quality. In the study by Saffan et al. [46], SeNPs (100–300 nm) foliar-applied to the tomato seedlings exposed to different kinds of low water quality, were capable of alleviating negative influence of oversalted water and boosting the tomato growth.

Consequently, the chemically synthesized SeNPs effect on plant growth depended upon the NP concentration and groundnut cultivar. On the other side, SeNPs at lower levels improve the activity of antioxidant enzymes [134]; thus, the optimization of doses and modes of SeNPs’ application is required before scaling up to the crop plant systems. Selenium remains the essential element required to support plant development as micronutrient until a Se concentration has reached up to the threshold level, then Se becomes toxic to plants [135]. Since SeNPs are capable of forming conjugates with distinct substances and undergo various physicochemical transforming processes in biosystem [56], SeNPs’ fate and feasible threat to living organisms should be reasonably considered.

## 8. Conclusive Remarks

Nanotechnology and nanomaterial production are growing exponentially. Advances in these fields provide decisive evidence for nanostructure usage in plant science, improving plant development and productivity, specifically in stressful environments. Selenium nanoparticles and their intricate forms exhibit promising potentialities of mitigating the adverse effects of abiotic stress on many crops. Innovative nano-assisted approaches have to be developed to take this opportunity provided by SeNPs. Nevertheless, the impact of SeNPs on humans and the detection of the acceptable limits are necessary. Even though the efficiency of SeNPs in abiotic-stress tolerance attracts much attention from researchers, a vast majority of these investigations are still at the laboratory step. Future studies should concentrate on designing SeNPs conjugated with the affordable, ecologically safe, and self-degradable (bio)matrix in order to transfer Se-containing nanomaterials to the field for sustained agricultural production.

## Figures and Tables

**Figure 1 ijms-26-01697-f001:**
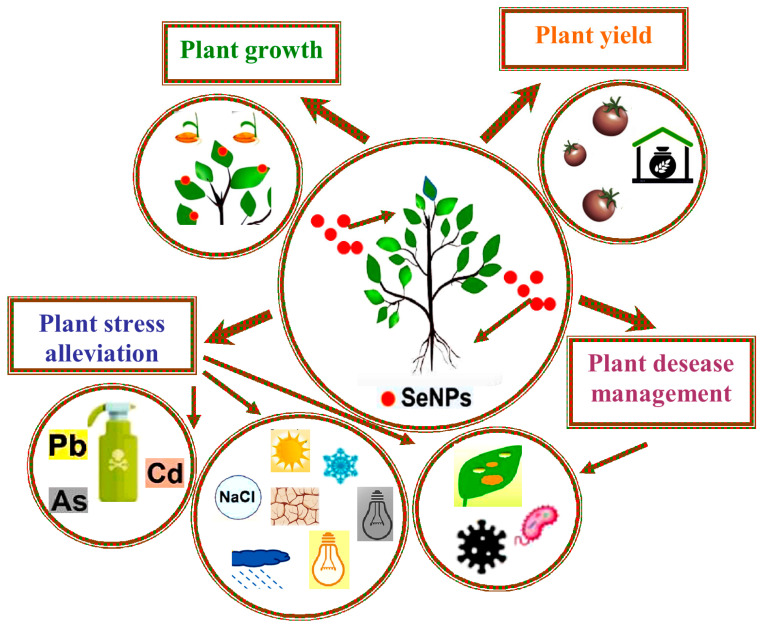
Schematic representation of principal lines of enquiry concerned SeNPs in plants. The extensive studies on feasible SeNPs involvement in plant research and development pay chief attention to the context of growth, yield, abiotic stress, and biotically caused deteriorations.

**Figure 2 ijms-26-01697-f002:**
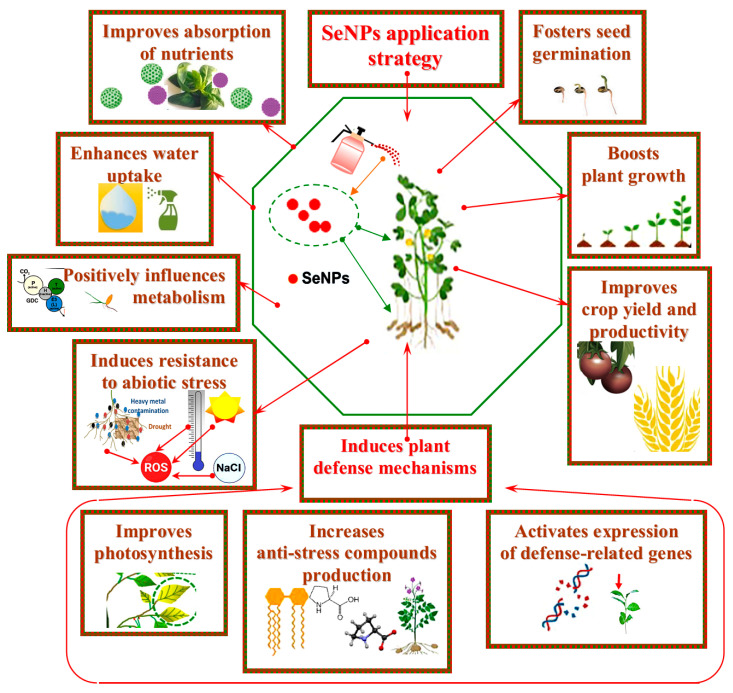
Graphical overview of the modifications caused by SeNP application strategy in plants. This strategy fosters seed germination and plant growth, exerts improving effects on basic and secondary metabolism, boosts resilience against abiotic stress by improving plant photosynthesis, increasing stress-mitigating compounds, thus inducing plant defense mechanisms. SeNP application fosters crop improvement and storage capacity.

**Table 1 ijms-26-01697-t001:** Research papers dealt with the agro-beneficial Se nanomaterials as cited by reviews since 2020.

Ref. (This Work)	Research Paper (Author-Year Citation)	Cited in Review (Author-Year Citation)	Review Essence
N.a. *	N.a.	Kamran et al., 2020 [8]	An overview of hazardous impacts of soil salinity in crops, tolerance mechanisms, and amelioration through selenium supplementation.
N.a.	N.a.	Salama et al., 2021 [9]	Applications of nanotechnology in vegetable crops.
[10]	Hussain et al., 2020	Rajput et al., 2021 [11]	Coping with the challenges of abiotic stress in plants: new dimensions in the field application of nanoparticles.
[12]	Zahedi et al., 2020		
[13]	Hernández-Hernández et al., 2019		
N.a.	N.a.	Ur Rahim et al., 2021 [14]	Nano-enabled materials promoting sustainability and resilience in modern agriculture.
[15]	Haghighi et al., 2014	Ali et al., 2021 [16]	Uptake, translocation, and consequences of nanomaterials on plant growth and stress adaptation.
[10]	Hussain et al., 2020	Avellan et al., 2021 [17]	Role of inorganic nanoparticle properties on their foliar uptake and in planta translocation. Highlights the material design opportunities and the knowledge gaps for targeted, stimuli driven deliveries of safe nanomaterials for agriculture.
N.a.	N.a.	El-Saadony et al., 2022 [18]	Role of nanoparticles in enhancing crop tolerance to abiotic stress: a comprehensive review.
[15]	Haghighi et al., 2014	Javed et al., 2022 [19]	Nanotechnology for endorsing abiotic stress: a review on the role of nanoparticles and nanocompositions.
[13]	Hernández-Hernández et al., 2019		
[20]	El-Saadony et al., 2021	Khalid & Iqbal Khan, 2022 [21]	Nanoparticles: the plant savior under abiotic stress.
[22]	Djanaguiraman et al., 2018		
[23]	González-García et al., 2021		
[12]	Zahedi et al., 2020	Kandhol et al., 2022 [24]	Nanoparticles as potential hallmarks of drought stress tolerance in plants.
[25]	Zahedi et al., 2021		
[26]	Ikram et al., 2020		
N.a.	N.a.	Salem et al., 2022 [27]	A comprehensive review of nanomaterials: types, synthesis, characterization, and applications. Describes the processes for green approaches to make nanomaterials of metals, metal oxides, graphene oxides, and polymers.
[28]	Badawy et al., 2021	Aguirre-Becerra et al., 2022 [29]	Nanomaterials as an alternative to increase plant resistance to abiotic stress.
[30]	Siddiqui et al., 2021	Sarkar et al., 2022 [31]	Recent advances in nanomaterial-based sustainable agriculture.
[25]	Zahedi et al., 2021	Rasheed et al., 2022 [32]	The role of nanoparticles in plant biochemical, physiological, and molecular responses under drought stress.
[33]	Zahedi et al., 2019		
[12]	Zahedi et al., 2020		
[34]	Seliem et al., 2020	Liu et al., 2022 [35]	Selenium regulates antioxidants, photosynthesis, and cell permeability in plants under various abiotic stresses.
[36]	Sardar et al., 2022		
[37]	Sheikhalipour et al., 2021		
N.a.	N.a.	Periakaruppan et al., 2023 [2]	New perception about the use of nanofungicides in sustainable agriculture practices.
N.a.	N.a.	Hayat et al., 2023 [38]	Nanoparticles and their potential role in plant adaptation to abiotic stress in horticultural crops.
N.a.	N.a.	Shelke et al., 2023 [39]	Enhancing secondary metabolites and alleviating environmental stress in crops with mycogenic nanoparticles: a comprehensive review.
N.a.	N.a.	Samynathan et al., 2023 [40]	Tthe impact of nano-Se on plant growth, metabolism, and stress tolerance. The approach to SeNP synthesis using plant extracts.
[25]	Zahedi et al.; 2021	Chandrashekar et al., 2023 [41]	Nanoparticle-mediated amelioration of drought stress in plants: a systematic review.
[42]	Gholami et al., 2022		
[26]	Ikram et al., 2020		
[20]	El-Saadony et al., 2021		
[25]	Zahedi et al., 2021	Al-Khayri et al., 2023 [43]	The role of nanoparticles in response of plants to abiotic stress at physiological, biochemical, and molecular levels.
[44]	Wang et al., 2021		
[22]	Djanaguiraman et al., 2018		
[22]	Djanaguiraman et al., 2018	Nawaz et al., 2023 [45]	Nanobiotechnology in crop stress management: an overview of novel applications.
[34]	Seliem et al., 2020		
[46]	Saffan et al., 2022		
[15]	Haghighi et al., 2014		
[47]	Nagdalian et al., 2023	Devi et al., 2023 [48]	Selenium nanomaterial is a promising nanotechnology for biomedical and environmental remediation.
[49]	Sheikhalipour et al., 2023		
[50]	Azimi et al., 2021	Zhang et al., 2023 [51]	Recent research progress on the synthesis and biological effects of selenium nanoparticles.
[52]	Li et al., 2022		
N.a.	N.a.	Ain et al., 2024 [53]	Deciphering the role of nanoparticles in stimulating drought and salinity tolerance in plants.
N.a.	N.a.	Yadav et al., 2024 [6]	Nanoparticle-mediated defense priming: a review of strategies for enhancing plant resilience against biotic and abiotic stress.

* N.a. means “not applicable”.

## Data Availability

All data are available within the manuscript.

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
