# Peer review of "Selenium-Containing Nanoformulations Capable of Alleviating Abiotic Stress in Plants"

_ijms, 2025, doi:10.3390/ijms26041697_

Round 1
Reviewer 1 Report
Comments and Suggestions for Authors
Detail Comments
1. The manuscript entitled as “Selenium-containing nanoformulations alleviating abiotic plant stress” is a review article which discusses about the importance of Nanomaterials in improving plant growth and enhance crop productivity.
2. The author discussed the importance of Selenium (Se). Selenium (Se) is put into the category of beneficial elements in plants. Se- enriched crops serve as a good dietary resource for Se-supplemented food and feed owing to their high bioavailability and accessibility. Researchers from different fields including plant science and nanoscience should better encourage possible innovative approaches dealt with abiotic stresses in agriculture. Predominating mechanisms of plants to ameliorate the stressful conditions are gener-ally considered to be assisted by Se nanoparticles (SeNPs) implementation. Increasing evidence of earlier research revealed that SeNPs could enhance plant growth and development, nutrient bioa- vailability, soil fertility, and stress response while maintaining the environment safe. Meanwhile some previous studies reported that SeNPs might have both positive and negative influence on plants mainly depending on the diverse Se-nanomaterial traits, its doses, and plant species. More attention is required to understand the SeNPs impact on crops under different abiotic stresses.
3. According to author in light of the current research demand for SeNPs and their long-term application in plants, the goal of this review is to focus on current beneficial SeNPs research trends, comparisons with bulk Se, insights into mechanisms of abiotic stress alleviation in plants, existing research uncertainties, and practical challenges for SeNPs applications under varying environment.
4. In summary, this manuscript has several positive attributes, but major improvements could further enhance the research.
Title
The title of the article “Selenium-containing nanoformulations alleviating abiotic plant stress” does not look quite attractive for the readers of this area. In addition it also look grammatically wrong. I suggest please change the title of the article.
1. Abstract
Line 8. “Climate change causes various types of abiotic stress in plants, thus affecting plant”
Please replace “Climate change causes” with “Climate changes cause”
Line 12. Selenium (Se) is put into the category of beneficial elements in plants.
Please revise this sentence such as “Selenium (Se) has been put into the category of beneficial elements in plants”.
Line 15. “nanoscience should better encourage possible innovative approaches dealt with abiotic stresses in”
Please revise this sentence and replace “dealt” to “that are linked with”
Line 16. Predominating mechanisms of plants to ameliorate the stressful conditions are generally considered to be assisted by.
This sentence needs to be revised and it can be written as “implementation of Se nanoparticles (SeNPs) is considered as one of the predominating mechanisms by plants to ameliorate the stressful conditions.
Introduction
Line 36 “increased interest” should be replaced by “more interest”
Line 43 “Highly relevant is a necessity to develop deeper insight into status of Se nanoparticles (SeNPs) in plant research”
Please revise this sentence. The meaning does not look clear.
Heading 2 “Recent reviews dealt with abiotic stress in plants exposed to NPs: Novelty statement”
From Line 62 to Line 177 have so much information on other atur names. The presentation style is not quite attractive and reliable. This is a review article and author should present the ideas in ascientific way. Although this section looks like a review of Literature from a PhD thesis section. Please revise the above mentions lines thoroughly and present the other scientists’ findings with general statements.
Line 178 -180
“Although several excellent investigations have been done on the NPs-induced stress-tolerance mechanisms in various crops (Table 1), there is no comprehensive review on the Se-containing nanostructures-mediated improvements in plants”, This sentence should be revised such as “Although several excellent investigations have been done on the NPs-induced stress-tolerance mechanisms in various crops (Table 1), there is no comprehensive review has been done so far on the Se-containing nanostructures-mediated improvements in plants”.
Line 180-184 “We follow a necessity to elucidate the role of Se, bulk form compared to SeNPs, in the improvement of common morpho-physiological and molecular responses of various plants subjected to abiotic stress, with the intention of identifying opportunities to conduct further research”.
Please revise this text. Better to write these lines in passive voice since this is a review article.
“We follow”, there is only one author found in this article so why we follow?
Heading 3 “Effects of exogenous bulk selenium on plant growth and redox characters”
Although the information provided in this section is quite useful and important but I suggest that authors should carefully revise the English grammar and typo errors where it is needed.
In addition, the similarity report for overall text is too high that is about 46%. Kindly reduce the similarity report along with the revision of the text.
Comments on the Quality of English Language
1. Abstract
Line 8. “Climate change causes various types of abiotic stress in plants, thus affecting plant”
Please replace “Climate change causes” with “Climate changes cause”
Line 12. Selenium (Se) is put into the category of beneficial elements in plants.
Please revise this sentence such as “Selenium (Se) has been put into the category of beneficial elements in plants”.
Line 15. “nanoscience should better encourage possible innovative approaches dealt with abiotic stresses in”
Please revise this sentence and replace “dealt” to “that are linked with”
Line 16. Predominating mechanisms of plants to ameliorate the stressful conditions are generally considered to be assisted by.
This sentence needs to be revised and it can be written as “implementation of Se nanoparticles (SeNPs) is considered as one of the predominating mechanisms by plants to ameliorate the stressful conditions.
Introduction
Line 36 “increased interest” should be replaced by “more interest”
Line 43 “Highly relevant is a necessity to develop deeper insight into status of Se nanoparticles (SeNPs) in plant research”
Please revise this sentence. The meaning does not look clear.
Heading 2 “Recent reviews dealt with abiotic stress in plants exposed to NPs: Novelty statement”
From Line 62 to Line 177 have so much information on other atur names. The presentation style is not quite attractive and reliable. This is a review article and author should present the ideas in ascientific way. Although this section looks like a review of Literature from a PhD thesis section. Please revise the above mentions lines thoroughly and present the other scientists’ findings with general statements.
Line 178 -180
“Although several excellent investigations have been done on the NPs-induced stress-tolerance mechanisms in various crops (Table 1), there is no comprehensive review on the Se-containing nanostructures-mediated improvements in plants”, This sentence should be revised such as “Although several excellent investigations have been done on the NPs-induced stress-tolerance mechanisms in various crops (Table 1), there is no comprehensive review has been done so far on the Se-containing nanostructures-mediated improvements in plants”.
Line 180-184 “We follow a necessity to elucidate the role of Se, bulk form compared to SeNPs, in the improvement of common morpho-physiological and molecular responses of various plants subjected to abiotic stress, with the intention of identifying opportunities to conduct further research”.
Please revise this text. Better to write these lines in passive voice since this is a review article.
“We follow”, there is only one author found in this article so why we follow?
Heading 3 “Effects of exogenous bulk selenium on plant growth and redox characters”
Although the information provided in this section is quite useful and important but I suggest that authors should carefully revise the English grammar and typo errors where it is needed.
Reviewer 2 Report
Comments and Suggestions for Authors
The manuscript provides a comprehensive review on selenium nanoparticles (SeNPs) as a promising solution to mitigate the impacts of abiotic stress, such as reduced plant growth and crop yields. SeNPs have demonstrated potential to enhance plant growth, nutrient bioavailability, soil fertility, and stress tolerance while ensuring environmental sustainability. However, their effects vary depending on nanoparticle traits, application doses, and plant species, with both beneficial and adverse outcomes reported. The manuscript references various studies addressing these uncertainties and practical challenges. It also examines current research trends, mechanisms of stress alleviation, comparisons with bulk selenium, and the potential of SeNPs across diverse environmental conditions. The review is comprised of various literature sources, over which 60% are from the past 5 years, which emphasizes the importance of the research topic.
The primary gap of knowledge identified in this review is the variability in the effects of SeNPs on plants under stress. Even though there are multiple research articles on the topic, there was no broad review on the SeNPs effect on plants in the light of their physiological, biochemical and molecular mechanisms of stress alleviation, and their interactions with diverse environmental conditions and potential strategies to increase productivity.
Table and figures are very well presented and highlight the main points of the work.
The abstract is longer than 200 words.
I believe that references in the text should be more like ref. 68 on line 390 rather than those on lines 65,75, 83, 93, 103, 105, 107, 116,119,120, 123, etc.
Line 71- salt?
There should be a separate chapter discussion according to the instructions
You should broaden a bit the Conclusions section and add Chapter Future directions, starting with the information after line 840.
Round 2
Reviewer 1 Report
Comments and Suggestions for Authors
Thank you very much for thoroughly revision of the article and point to point addressing the comments.
All the comments have been carefully revised but I am still wondering about the title of the revised manuscript. Since it has ben mentioned in the responses the title has been changed but I am sorry I could not find the revised title in the revised version.
Please pay attention on the revision of the title.
Other revisions are fine and after revising the title the manuscript can be accepted for publication.
Author Response
Dear Reviewer,
Thank you once again for handling and thorough reviewing the manuscript.
Comments and Suggestions for Authors
(a) All the comments have been carefully revised but I am still wondering about the title of the revised manuscript. Since it has ben mentioned in the responses the title has been changed but I am sorry I could not find the revised title in the revised version.
Please pay attention on the revision of the title.
Other revisions are fine and after revising the title the manuscript can be accepted for publication.
Reply (a): Sorry for the unfortunate mistake, now I use the modified title, and changes are highlighted in dark-blue font in this version.
Sincerely,
Dr. Olga M. Tsivileva
Laboratory of Microbiology, Institute of Biochemistry
and Physiology of Plants and Microorganisms,
Saratov Scientific Centre of the Russian Academy of Sciences,
13 Prospekt Entuziastov, Saratov 410049, Russia
tsivileva_o@ibppm.ru